# A First Assessment of Carbon Nanotubes Grown on Oil-Well Cement via Chemical Vapor Deposition

**DOI:** 10.3390/nano12142346

**Published:** 2022-07-09

**Authors:** Luca Lavagna, Mattia Bartoli, Simone Musso, Daniel Suarez-Riera, Alberto Tagliaferro, Matteo Pavese

**Affiliations:** 1Department of Applied Science and Technology, Politecnico di Torino, C.so Duca degli Abruzzi 24, 10129 Turin, Italy; simone.musso@polito.it (S.M.); alberto.tagliaferro@polito.it (A.T.); matteo.pavese@polito.it (M.P.); 2National Interuniversity Consortium of Materials Science and Technology (INSTM), Via G. Giusti 9, 50121 Florence, Italy; mattia.bartoli@iit.it; 3Center for Sustainable Future Technologies @POLITO, Istituto Italiano di Tecnologia, Via Livorno 60, 10144 Turin, Italy; 4Massachusetts Institute of Technology, Department of Civil and Environmental Engineering, 77 Massachsetts Ave., Cambridge, MA 02139, USA; 5Department of Structural, Geotechnical and Building Engineering, Politecnico di Torino, C.so Duca Degli Abruzzi 24, 10129 Turin, Italy; daniel.suarez@polito.it; 6Faculty of Science, Ontario Tech University, 2000 Simcoe Street North, Oshawa, ON L1G 0C5, Canada

**Keywords:** carbon nanotubes, cement clinker, chemical vapor deposition, composites

## Abstract

In this study, carbon nanotubes (CNTs) were synthesized on an oil-well cement substrate using the chemical vapor deposition (CVD) method. The effect of synthesis process on cement was investigated in depth. In this regard, FE-SEM, RAMAN and X-Ray spectroscopy were used to characterize the cement before and after the synthesis process to reveal the modifications to the cementitious matrix and some unique morphological features of CNTs.

## 1. Introduction

Since the first observation [1] and report of multi-walled carbon nanotubes (MWCNTs) in 1991 [2], and the observation of single-walled carbon nanotubes (SWCNTs) in 1993 [3], carbon nanotubes (CNTs) have attracted the attention of scientific and industrial world. CNTs have unique chemical and physical properties [4] that allow for their use in several applications [5,6,7,8]. The most common way of synthesizing carbon nanotubes is chemical vapor deposition (CVD) [9]. CVD allows, in addition to controlling the quality and morphology of CNTs [10,11], for nanotubes to grow on different types of supports [12,13]. In the last decade, the use of CNTs in cement- and concrete-based composites has received particular attention [14]. CNTs are particularly used to reinforce structures and enable the real-time monitoring of structures thanks to their conductive and piezoresistive nature [15,16,17]. The main problem with their use in cement is the difficulty of dispersion [17] and interaction with the cement [18]. Several works in the literature report different solutions to improve these aspects such as chemical functionalization and sonication [19,20,21,22,23,24]. However, these techniques are destructive and often do not guarantee a good interaction and dispersion in the matrix [25,26,27,28]. Other work has used CVD synthesis methods for nanotube growth on Portland cement [29] and, in one case, on class G cement [30]; however, the method and temperature used for the synthesis of CNTs differed to those used in this study. In this work, we report a new, alternative method to properly disperse CNTs in cement matrix by directly growing, via CVD, CNTs on a cement clinker (cement powder).

## 2. Materials and Methods

The cement used in this study is an American Petroleum Institute (API) oil-well cement Class G (Lafarge North America, Reston, VA, USA). Camphor (purity > 99%) and ferrocene (purity > 99%) were purchased by Sigma-Aldrich (Darmstadt, Germany) and used without any further purification for CNT production. CVD process was run accordingly with that reported by Musso et al. [31], using camphor and ferrocene with a ratio of 20:1 by weight and a camphor/cement ratio of 1 by weight. Cement clinker was placed inside the reactor and heated at 1000 °C; once the temperature was reached, a flux of argon was used to carry the vapors of the camphor/ferrocene mixture into the reactor, as shown in Figure 1.

X-ray diffraction (XRD) patterns were obtained using the X-ray diffractometer PW3040/60 X’Pert PRO MPD from, Panalytical BV, Almelo, Netherlands in a Bragg–Brentano geometry, with Cu Kα anode source at 40 KV and 40 mA.

Raman spectra were collected using a Renishaw inVia (H43662 model, Gloucestershire, UK) equipped with a green laser line (514 nm) with a 50× objective. Raman spectra were recorded in the range from 250 cm^–1^ to 3500 cm^–1^.

CNTs containing cement morphology was evaluated using a field emission scanning electrical microscope (FE-SEM, Zeis SupraTM40, Oberkochen, Germany).

## 3. Results and Discussion

### 3.1. Assessment Analysis of CNTs Based Cement Composites

The cement clinker recovered from the CVD process was preliminarily analysed though Raman spectroscopy [32] (Figure 2) to confirm the formation of CNTs.

The raman spectra of cement showed an inhomogeneous composition, as reported by Deng et al. [33], with grains rich in several phases of aluminosilicate (C_3_S, C_2_S, C_3_A) (Figure 2a) and grains rich in gypsum (Figure 2b). After CVD, the Raman spectra (Figure 1d) showed the I_D_ and I_G_ peaks, centred at 1346 cm^−1^ and 1575 cm^−1^, respectively, proving the formation of CNTs. The high temperature of the process led to a quite good-quality MWCNTs with an I_D_/I_G_ of 0.98 [34]. It is worth noting that both the annealed clinker (Figure 2c) and the CVD-treated version (Figure 2d) exhibited a similar thermal conversion of CaSO_4_·2H_2_O into CaO and SO_2,_ according to the thermal degradative mechanism reported by West et al. [35]. Moreover, for the CVD-treated clinker, the presence of iron precursors might promote and accelerate the gypsum thermal degradation.

As shown in Figure 3, CNTs grown on cement clinker were analysed by FE-SEM to evaluate their morphology.

As reported in Figure 3a, CNTs grew on the grains of clinker without forming any appreciable bundle, in accordance with with the results reported by Ghaharpour et al. [36], who used ethylene as precursor. Two main typologies of CNTs could be detected by FE-SEM analysis, as shown in Figure 3b,c. Some CNTs appear to have a smoother surface combined with a smaller average diameter (100 nm square circled), while others have a larger average diameter (approximately 200 nm) and seem to made by an inner CNT of approximately 10–50 nm in diameter, covered by a multilayered structure of approximately 50 nm in thickness (red circled). The cross-sections of such structures are highlighted with red circles. We hypothesized (see Appendix A) that, during the growth of the CNTs, their external walls are incrementally covered by layers of iron and iron/carbon compounds generated in excess as by-products of the CVD process.

### 3.2. X-ray Diffraction Analysis

The XRD spectra taken on pure cement clinker, cement clinker annealed at 1000 °C, and after the growth of CNTs are reported in Figure 4. Their analysis shows a complex phase composition. In the pristine class G cement clinker, the presence of the main cement phases is evident: bicalcium silicate (C2S), tricalcium silicate (C3S), calcium ferroaluminate (C4AF), calcium sulfate dihydrate (gypsum). Distinguishing between C2S and C3S is not easy, since the peaks are often superimposed. It seems that the main phase between the two is C3S, with only a minor contribution of C2S.

As previously shown with Raman analyses, after heat treatment at 1000 °C the annealed clinker exhibits the decomposition of both gypsum and its insoluble phase (CaSO_4_ also called “insoluble anhydrite”) into calcium oxide and sulfur oxide. This phenomenon is confirmed by the appearance of the peaks in the decomposition product CaO. Moreover, probably due to exposure to ambient humidity after thermal treatment, it is possible that part or all of the CaO is hydrated to Ca(OH)_2_. Again, it seems that the C3S phase is present in a much larger fraction than C2S.

On the other hand, the clinker on which CNTs were grown via CVD shows several different phase modifications with respect to both the pristine and the 1000 °C-annealed class G cement. First, even though gypsum and its insoluble phase are no longer observed, this phenomenon is different from the one described for the annealed clinker because the presence of carbon during the CVD process activates the catalytic reduction reaction of CaSO_4_ into CaS, as shown by Oh and Wheelock [37]. The C2S phase increases substantially, suggesting that the conversion of C3S into C2S and CaO is favored during the CVD growth of nanotubes, due to the catalytic role played by the iron used for the synthesis of CNTs [38]. This mechanism is also confirmed by the fact that CaO peaks in the CVD-treated clinker are much more intense than the ones detected for the annealed clinker [39]. As seen for the annealed clinker, the Ca(OH)_2_ peak increases as well, likely due to the water uptake after the synthesis. Finally, a small peak is observed at around 26.2°, corresponding to the main peak in CNTs, and another peak is observed at around 44.5°, corresponding to the main peak in α-Fe. The presence of both these peaks provides further confirmation of the observed CNT growth.

## 4. Conclusions

In this paper, we have demonstrated the direct growth in CNTs on cement clinker powder by CVD. Despite the fact that the process can cause the degradation of some cement phases, this route could be further explored to avoid the problems with the functionalization and dispersion of CNTs for cement-based composite applications.

## Figures and Tables

**Figure 1 nanomaterials-12-02346-f001:**
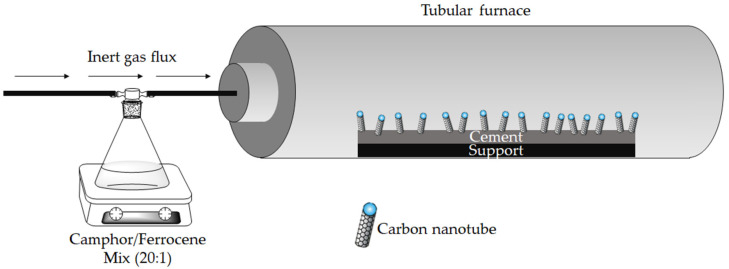
Schematic representation of the system used for carbon nanotube synthesis.

**Figure 2 nanomaterials-12-02346-f002:**
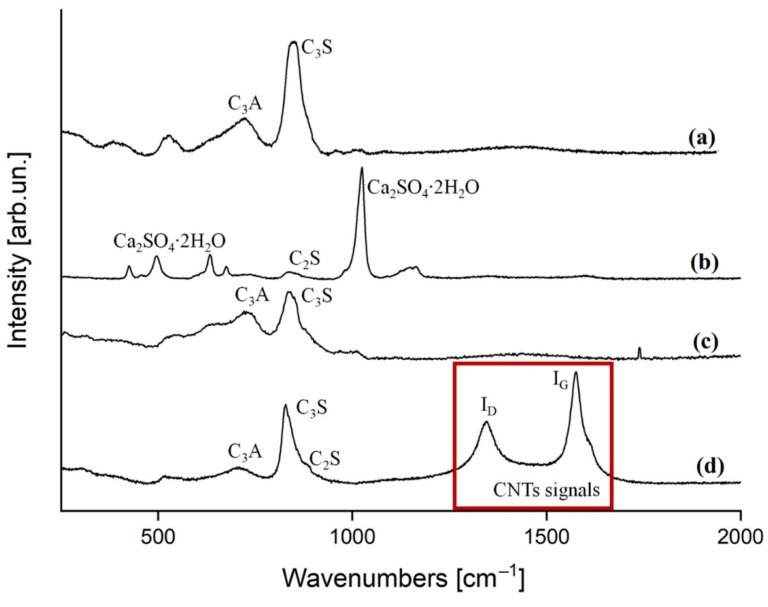
Raman spectra (250–2000 cm^−1^ range) collected on (**a**) pristine cement clinker rich in aluminium silicate, (**b**) pristine cement clinker rich in calcium sulphate dihydrate (gypsum), (**c**) cement annealed at 1000 °C, and (**d**) cement after CVD growth.

**Figure 3 nanomaterials-12-02346-f003:**
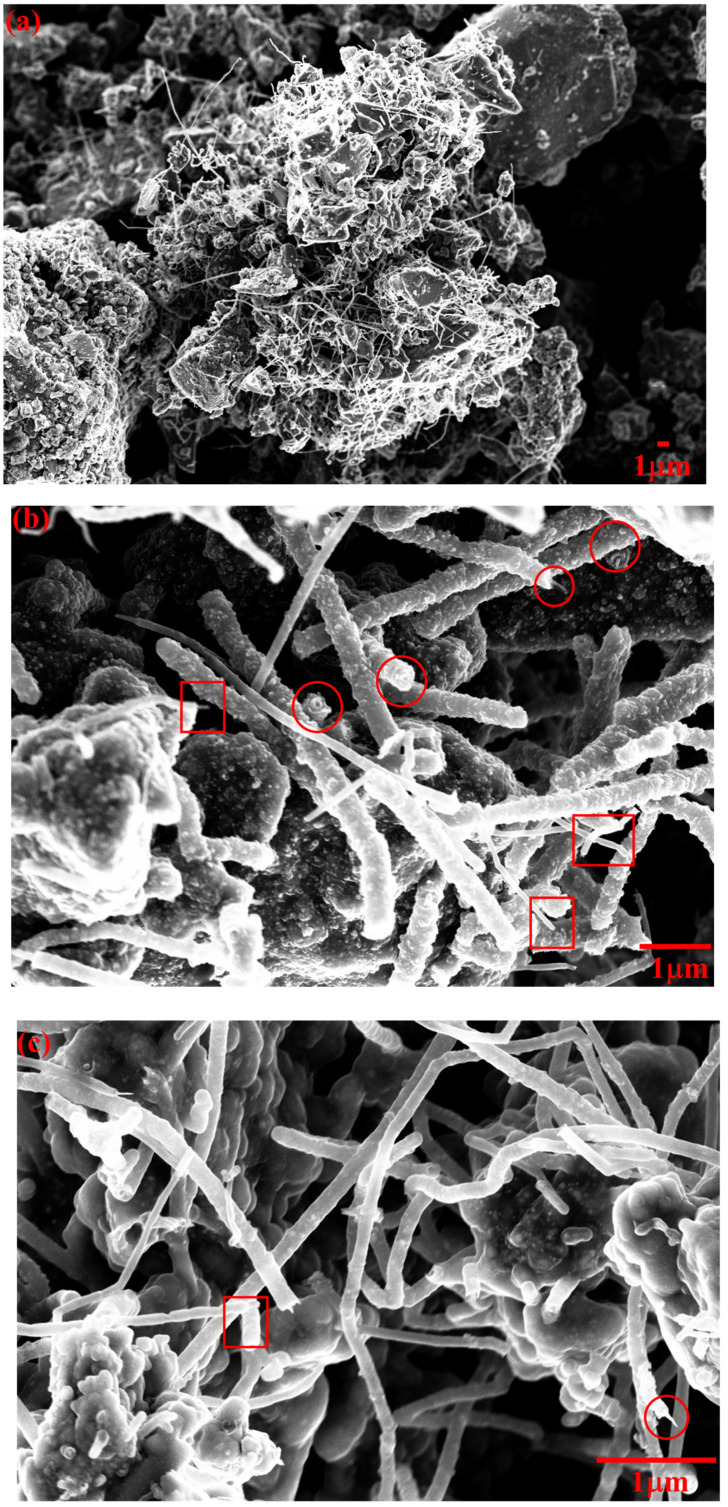
FE-SEM images of CNTs grown on cement clinker, with magnification of (**a**) 3000, (**b**) 30,000 and (**c**) 50,000 times. Cross sections of individual CNTs are red circled/squared.

**Figure 4 nanomaterials-12-02346-f004:**
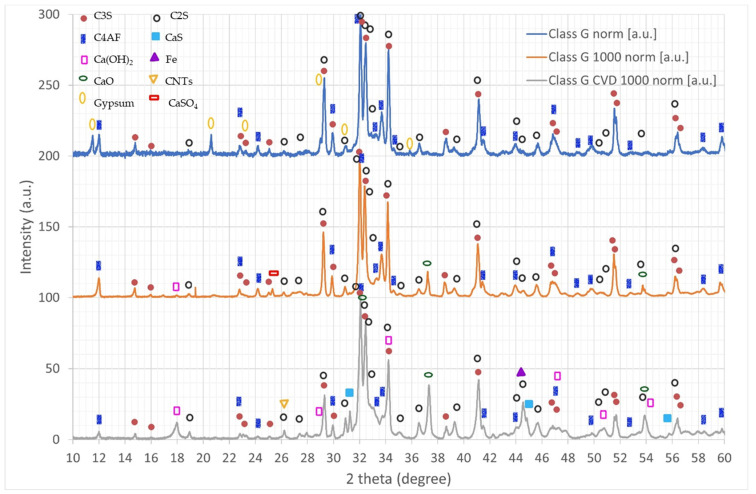
X-ray diffraction spectra of pristine clinker (**top**), clinker annealed at 1000 °C (**middle**), and clinker CVD-treated for grown CNTs (**bottom**).

## Data Availability

The data presented in this study are available on request from the corresponding author.

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
