# Peer review of "A First Assessment of Carbon Nanotubes Grown on Oil-Well Cement via Chemical Vapor Deposition"

_nanomaterials, 2022, doi:10.3390/nano12142346_

Round 1

Reviewer 1 Report

The authors investigated the carbon nanotubes containing cement made by annealing the cement clinker at 1000 oC using camphor and ferrocene as the carbon source and catalyst. It was regarded as a new alternative method to disperse CNTs in cement matrix by directly growing, via CVD, CNTs on cement clinker. After checking throughout this manuscript, it is recommended that it is with a major revision before consideration for acceptance.

Please indicate the process by a detailed diagram (including the flux of the gas Argon) where camphor and ferrocene are precisely placed.

Please supplement the EDX elemental mapping of so-called CNTs to analyze external wall differences and support the discussion by the authors.

Reviewer 2 Report

The communication “A first assessment of CNTs grown on oil-well cement via CVD” by Luca Lavagna, Mattia Bartoli, Simone Musso, Daniel Suarez-Riera, Alberto Tagliaferro, Matteo Pavese, presents new results with regard of synthesis process of carbon nanotubes (CNTs) on oil-well cement substrate using chemical vapor deposition (CVD) method.

First of all, in the title of the communication carbon nanotubes and chemical vapor deposition appear in abbreviation, CNT and CDV respectively, but in the abstract the explanation of these terms are given. I suggest writing in the title unabbreviated expressions.

Lines 29, 31, 61, 66, 70, 72, 111, 114, 116. The lack of space before brackets with references.

Line 37 “…however, the method and temperature used for the synthesis are different”. The method different from which one, that of the present study? I recommend rewriting this sentence in clearer way.

Line 39. I would like to recommend to authors to explain the term “cement clinker” more detailed.

Lines 60 – 70The high temperature of the process led to a quite good quality MWCNTs with an ID/IG of 0.98”. How the fact that multi-walled carbon nanotubes (MWCNTs) and not single-walled carbon nanotubes (SWCNTs) were synthesized, was proven? And how it was concluded a good quality of the MWCNTs? I would rather not use “quite” and try to specify it. In general way, it is not clear from the text what CNTs the study is dealing with, SWCNTs, MWCNTs or both kinds of the nanotubes.

Line 70. Instead of “Interestingly” I suggest using “It is worth to be noted”.

Lines 81-82 In “…according with the results reported by Ghaharpour et al. [36], where ethylene was used as precursor” I suggest writing “…[36], who used ethylene as precursor”.

Line 85 “circa 200 nm” means “at about” or “approximately?

Lines 104 – 105 “Again, it seems that the 104 C3S phase is present in a much larger fraction than C2S”. Is it possible to quantify these fractures?

Lines 117 – 119 “Finally, a small peak related to CNTs is present together with a peak attributed to α-Fe, confirming the observed CNTs growth”. The fact that a peak indicates growth of the CNTs should be explained in a clearer way.

Reviewer 3 Report

This paper reports the synthesis of carbon nanotubes on cement substrate by CVD. This paper emphasizes the importance of properly dispersing CNT in cement matrix; however, this paper does not show appropriate discussion about the dispersion. In addition, this paper show conventional technique to make carbon nanotubes without a novel strategy. I consider this paper does not include the clear scientific significance and new findings for communication articles on nanomaterials. Therefore, I can not recommend publishing this paper in this journal.

Round 2

Reviewer 1 Report

I suggest it be acceptable for consideration for publication.

Author Response

We thank the reviewer for the positive comment

Reviewer 2 Report

The manuscript can be published now.

Author Response

(The authors gave the same response as above.)

Reviewer 3 Report

I consider the authors answered comments adequately on the referee’s criticism. I feel this paper seems to have an original strategy. My minor comments are listed below.

1)    How about the growth yield of carbon nanotubes?

2)    How about the structures and dispersity of carbon nanotubes inside or outside the cement? 
